# Analytical Modeling and Optimization of Cu_2_ZnSn(S,Se)_4_ Solar Cells with the Use of Quantum Wells under the Radiative Limit

**DOI:** 10.3390/nano13142058

**Published:** 2023-07-12

**Authors:** Karina G. Rodriguez-Osorio, Juan P. Morán-Lázaro, Miguel Ojeda-Martínez, Isaac Montoya De Los Santos, Nassima El Ouarie, El Mustapha Feddi, Laura M. Pérez, David Laroze, Soumyaranjan Routray, Fernando J. Sánchez-Rodríguez, Maykel Courel

**Affiliations:** 1Centro Universitario de los Valles, Universidad de Guadalajara, Carretera Guadalajara—Ameca Km. 45.5, Ameca C.P. 46600, Jalisco, Mexico; karina.osorio@academicos.udg.mx (K.G.R.-O.); pablo.moran@academicos.udg.mx (J.P.M.-L.); miguel.ojeda9380@academicos.udg.mx (M.O.-M.); 2Instituto de Estudios de la Energía, Universidad del Istmo, Santo Domingo Tehuantepec C.P. 70760, Oaxaca, Mexico; isaacms88@gmail.com; 3Group of Optoelectronic of Semiconductors and Nanomaterials, ENSAM, Mohammed V University in Rabat, Rabat 10100, Morocco; nassima.elouari@usmba.ac.ma (N.E.O.); e.feddi@um5r.ac.ma (E.M.F.); 4Institute of Applied Physics, Mohammed VI Polytechnic University, Lot 660, Hay Moulay Rachid, Ben Guerir 43150, Morocco; 5Departamento de Física, FACI, Universidad de Tarapacá, Casilla 7D, Arica 1000000, Chile; lperez@uta.cl; 6Instituto de Alta Investigación, Universidad de Tarapacá, Casilla 7D, Arica 1000000, Chile; dlarozen@uta.cl; 7Department of Electronics and Communication Engineering, SRM Institute of Science and Technology, Chennai 603203, India; s.r.routray@ieee.org; 8Facultad de Ciencias Físico-Matemáticas, Universidad Autónoma de Sinaloa, Culiacán C.P. 80010, Sinaloa, Mexico; sanchezr@uas.edu.mx

**Keywords:** kesterite solar cells, quantum wells, solar cell modeling, radiative limit

## Abstract

In this work, we present a theoretical study on the use of Cu_2_ZnSn(S,Se)_4_ quantum wells in Cu_2_ZnSnS_4_ solar cells to enhance device efficiency. The role of different well thickness, number, and S/(S + Se) composition values is evaluated. The physical mechanisms governing the optoelectronic parameters are analyzed. The behavior of solar cells based on Cu_2_ZnSn(S,Se)_4_ without quantum wells is also considered for comparison. Cu_2_ZnSn(S,Se)_4_ quantum wells with a thickness lower than 50 nm present the formation of discretized eigenstates which play a fundamental role in absorption and recombination processes. Results show that well thickness plays a more important role than well number. We found that the use of wells with thicknesses higher than 20 nm allow for better efficiencies than those obtained for a device without nanostructures. A record efficiency of 37.5% is achieved when 36 wells with a width of 50 nm are used, considering an S/(S + Se) well compositional ratio of 0.25.

## 1. Introduction

The second generation of photovoltaics has received a lot of attention from the scientific community because of CdTe and CuInGaSe_2_ (CIGS) materials, with device efficiencies bigger than 22% [1]. Nevertheless, the low abundance of In, Ga, and Te along with Cd toxicity has limited the scale-up of this technology [2,3]. In order to solve these drawbacks, new materials based on non-toxic and earth-abundant elements have been extensively studied. Among the different proposals, the kesterite family based on Cu_2_ZnSnSe_4_ (CZTSe), Cu_2_ZnSnS_4_ (CZTS), and Cu_2_ZnSn(S,Se)_4_ (CZTSSe) constitutes potential materials for solar cell applications. In particular, they have been described as non-toxic and earth-abundant elements along with producing direct band-gap transitions that result in a high absorption coefficient [3]. Furthermore, an interesting feature of CZTSSe compound is its band-gap variation from 1.0 (CZTSe) to 1.5 eV (CZTS) with S/(S + Se) compositional ratio changing from 0 to 1 [4]. Although CZTSSe is an attractive material for device application, low efficiencies have been reported so far, with a recent certified record efficiency of 13.8% [5]. An important problem of CZTSSe solar cells in comparison to technologies such as CIGS is the deficit of open-circuit voltage (Voc) [3,6,7], while short-circuit current density (Jsc) values are close for both devices [8]. A high concentration of defects along with grain boundaries, defect clusters, and poor band-tailing characteristics have been identified as the main causes of Voc-deficit [9,10]. In particular, it was found that under Cu-poor and Zn-rich conditions, the relative concentration of the Zn_Cu_ and Sn_Cu_ defects and cluster defects such as [2Zn_Cu_ + Zn_Sn_] and [2Cu_Zn_ + Sn_Zn_] play a fundamental role in the Voc and thereby on the power conversion efficiency [9]. The relationships between structure, composition and electronic properties have been presented as a key factor for reducing the Voc-deficit in this technology [10]. However, despite efforts performed to improve the Voc parameter, so far, results on this research line have been discouraging. Consequently, new alternative approaches to improve CZTSSe solar cell efficiency are required.

As a general trend, the highest Voc in CZTSSe solar cells has been reported for devices based on CZTS material while the biggest Jsc values have been achieved when CZTSe is used as the absorber [3,7]. These results can be understood from the fundamentals of semiconductor physics, where lower band-gap materials as in the case of CZTSe can guarantee a higher absorption of photons and consequently a higher Jsc but low Voc, and vice versa. In order to solve these typical problems in photovoltaics, a new solar cell generation has emerged suggesting the use of novel devices that improve either Voc or Jsc in comparison to traditional devices from the second generation, which would result in an enhanced efficiency. One of these proposals considers tandem solar cells, where the device voltage is the sum of the different subcell output voltages, while current density is determined by the lowest one of subcells. Another interesting proposal concerns the use of nanostructures in solar cells to increase photon absorption, having an efficiency promotion if the Voc is not drastically reduced in comparison to the device without nanostructures [11,12,13,14]. That is, the use of nanostructures that allow lower energy photon absorption in solar cells would enhance Jsc in comparison to the reference device and thereby efficiency of the solar cell with nanostructures would be higher than that of the reference device if the Voc of both solar cells were similar.

Quantum wells are an attractive type of nanostructure with a great potential application into photovoltaics. The first proposal of quantum well solar cells was reported in 1990 [15]. Thereafter, different types of quantum well solar cells have been proposed and analyzed from experimental and theoretical studies [16,17,18,19,20,21,22]. A general feature of quantum wells is that absorption and recombination processes can be controlled through discrete eigenstates depending on well thickness and depth, allowing for finding the conditions that would result in an improved device performance. Quantum wells favor outspreading the quantum efficiency of the device. Therefore, a Jsc increase for solar cells with quantum wells is expected in comparison to the reference device. However, carriers at wells are confined, making their recombination likely, thereby reducing the output voltage of the device in comparison to the reference device. As a result, conditions that guarantee higher generation gain than recombination losses with the inclusion of wells should be studied. 

In CZTSSe solar cells, the construction of quantum well solar cells could be possible considering a type I heterostructure formation for CZTS/CZTSe, characterized by a continuous band-gap variation (1.0–1.5 eV) as a function of composition [4]. Since the highest open-circuit voltage in CZTSSe solar cells has been reported for CZTS solar cell [3,7], the use of CZTS as host material with quantum wells based on CZTSSe compound is expected to increase Jsc in comparison to the reference CZTS solar cell, with Voc values similar to the traditional CZTS solar cell resulting in a promoted efficiency. The first proposal of kesterite quantum well solar cells was presented in 2019 [23] and was further enriched by subsequent work, in which different types of materials belonging to the kesterite family were considered [24,25,26,27,28]. However, there are few works reported on this topic; all of which are in the early stages of development. In particular, more studies are required to attain complete knowledge of the physics governing kesterite solar cell with the inclusion of CZTSSe quantum wells, ranging from the study of eigenstate formation at wells to the role of well thickness, number, and composition on carrier transport. For this analysis, it is important to compare under the ideal conditions of the radiative limit results of solar cells with and without nanostructures. This would allow for a better understanding of the role of quantum wells in carrier transport and consequently on the device characteristics such as reverse dark current density (J_0_), short-circuit current density (Jsc), open-circuit voltage (Voc), and efficiency. So far, there is a lack of information in the literature on this type of study, which is imperative to understanding the viability of the proposal and to acquire more knowledge on the physics behind the use of kesterite quantum wells as well as conditions that allow for maximum solar cell performance. In this work, we analyze CZTS/CZTSSe quantum well solar cells, with a particular emphasis on finding conditions that allow for the best efficiencies while also aiming at understanding the physical mechanisms that govern this device through an analytical approach.

## 2. Theoretical Considerations

The analytical study of CZTSSe quantum well solar cells is carried out under the radiative limit in order to find conditions under which the maximum solar cell efficiencies are expected. This approach is only focused on band-to-band transitions, involving photon emission. Consequently, other losses due to photon reflection, device resistance, and defects at bulk and interfaces are neglected. This approach also considers that other materials composing the solar cell, such as the window layer, the buffer layer, and the contacts, present ideal properties such as good lattice matching and band alignment with no photon absorption as well as very long diffusion length values [29]. The purpose of using these ideal conditions for studying CZTSSe solar cells is to understand the physical mechanisms governing this type of device while at the same time finding conditions that would guarantee the best efficiency. The consideration of lack of bulk defects in CZTSSe implies the formation of an intrinsic material for barriers and wells. For the analysis, a bulk solar cell is first studied. This is an important step as it not only allows for finding the maximum efficiency expected without nanostructures for comparison with the ones obtained with quantum wells, but also sets the maximum Jsc and Voc values that can be attained for the bulk solar cell. The bulk CZTSSe solar cell behavior under the radiative limit can be evaluated by the current density–voltage characteristics (*J*-*V*), which are given by the following formula [29]:(1)$$J=qWn_{i}^{2}B\left[Exp\left(\frac{qV}{kT}\right)-1\right]-J_{PH}$$
where parameters such as absorber thickness (*W*), the radiative coefficient (*B*), intrinsic carrier concentration (*n_i_*), the thermal energy (*kT*), the terminal voltage (*V*), the photocurrent density (*J_PH_*), and electron charge (*q*) are required for calculations. Once conditions that optimize bulk solar cell are known, CZTS/CZTSSe quantum well solar cells are evaluated. Figure 1 presents a sketch diagram of a CZTS/CZTSe quantum well solar cell, for which there is the formation of wells for electrons and holes. In this particular example, CZTS constitutes the barrier material while CZTSe is the well material.

Under the radiative limit, the behavior of CZTSSe solar cells with quantum wells is estimated by the following formula [23,30]:(2)$$J_{QWSC}=qWn_{i}^{2}B\left\{1+f_{w}\left[\gamma_{B}\gamma_{DOS}^{2}\mathrm{Exp}\left(\frac{\Delta Eg-qFL_{w}}{kT}\right)-1\right]\right\}\left[Exp\left(\frac{qV}{kT}\right)-1\right]-J_{PH}$$
where, the first terms including absorber thickness (*W*), the radiative coefficient (*B*), and the intrinsic carrier concentration (*n_i_*) are the same as those defined in Equation (1), but now are used for barrier material, while terms within the curly brackets make reference to the increase in dark current density (*J*_0_) when using quantum wells. The other parameters required for calculations are the absorber material fraction filled with wells (*f_w_*), the ratio between the radiative coefficients corresponding to wells and barriers (*γ_B_*) and the well and barrier effective density of states ratio (*γ_DOS_*). Furthermore, Δ*Eg* stands for the barrier and well band-gap difference, $L_{w}$ is the well thickness, and *F* constitutes the electric field. Since quantum wells are often under the effect of an electric field due to the p-i-n structure, its presence should be assumed for calculations. The term *J_PH_* in Equation (2) is the photocurrent density when quantum wells are considered, which is evaluated by the following formula [23,30]:(3)$$J_{PH}=q\int F\left(\lambda \right)\left[1-\exp\left(-\alpha_{b}W-\alpha_{w}NL_{w}\right)\right]d\lambda$$
where the photon flux $F\left(\lambda \right)$ is taken from the AM1.5G solar spectrum (300 to 1300 nm), the absorption coefficients for bulk and well materials are *α_b_* and *α_w_*, respectively, and the well number is *N*. A straightforward expression is obtained for calculating the photocurrent density in solar cells without nanostructures (*J_PH_* in Equation (1)) when the number of wells in Equation (3) is set to zero.

To evaluate the behavior of CZTS solar cells with CZTSSe quantum wells, parameters such as absorption coefficient, density of states, and the radiative coefficient should be evaluated. Both recombination and absorption processes in quantum wells occur through discrete levels; consequently, it is important to find eigenstates that play important roles in quantum transitions within wells. Eigenvalues at wells—which would depend on well properties such as thickness and depth—are calculated from the Schrödinger equation while considering the boundary conditions of BenDaniel–Duke as well as the effective mass approach. Expressions available in the literature are used for calculating the bulk material and well absorption coefficients [29,31]. It is important to remark that for calculating the absorption coefficient, light hole-electron and heavy hole-electron transitions are considered for both bulk and well materials. In particular, for quantum wells, the absorption coefficient is a stepped function due to energy level discretization, which depends on well thickness and depth. Another important parameter to evaluate is the effective density of states at wells—a stepped function due to energy level discretization, depending on well thickness and depth. Taking into account the methodology discussed by Lade and Zahedi [32] as well as the Fermi–Dirac equation under the Maxwell–Boltzmann approximation, the electron and hole effective density of states at wells are given by the following formula [30]:(4)$$g_{e,h}=\frac{m_{e,h}^{*}kT}{\pi \hbar^{2}L_{w}}\sum_{i}\left[Exp\left(\frac{-E_{e,h,i}}{kT}\right)-Exp\left(\frac{-{∆E}_{c,v}}{kT}\right)\right]+g_{b}\left[2\sqrt{\frac{{∆E}_{c,v}}{\pi kT}}Exp\left(\frac{-{∆E}_{c,v}}{kT}\right)+Erfc\left(\sqrt{\frac{{∆E}_{c,v}}{kT}}\right)\right]$$
where, *m_e_*_,*h*_* is the electron and hole effective mass, ∆*E_c_*_,*v*_ is the conduction and valence band offsets for electrons and holes, respectively, the discretized energy levels at wells are given by *E_e_*_,*h*,*i*_, $g_{b}$ is the barrier effective density of states, and *Erfc* is the error function. The expression for calculating the electron and hole effective density of states in bulk can be found elsewhere. Once the values of electron, light hole, and heavy hole effective density of states ($g_{e}$, $g_{lh}$ and $g_{hh}$) are calculated, the well and barrier effective density of state can be found by the following formula:(5)$$g=\sqrt{g_{e}\left(g_{lh}+g_{hh}\right)}$$

The radiative coefficient (*B*) is another relevant parameter to find before evaluating Equations (1) and (2). The bulk and well radiative coefficients are calculated by the following equation [30]:(6)$$B=\frac{8\pi n_{r}^{2}}{c^{2}h^{3}n_{i}^{2}}\int_{E_{1}}^{\infty }\frac{\alpha E^{2}dE}{Exp\left(\frac{E}{kT}\right)-1}$$
where the Planck constant (*h*) and the speed of light (*c*) are fundamental parameters, while *n_r_* is the refraction index. *E*_1_ represents the minimum transition energy considered for calculations. That is, in the case of bulk material, *E*_1_ is the band-gap while in quantum wells it is the band-gap shifted by the sum of the hole and electron confinement energies (ground states). Furthermore, the intrinsic carrier concentration (*n_i_*) is calculated from the effective density of states for electrons and holes at wells and barriers, respectively, and *α* is the absorption coefficient—either for bulk or wells. Due to the 2-D density of states nature of the quantum wells, the absorption coefficient and, consequently, the integration presented in Equation (6), occurs in a continuous range but is stepped due to energy level discretization in one of the directions. The solar cell characteristics can be estimated from expressions (1) and (2), once radiative coefficient, effective density of states, absorption coefficient, and photocurrent density are evaluated. Thereafter, efficiency, Jsc, and Voc are calculated.

For calculations, information reported in the literature for band-gap variation (1.0–1.5 eV) [4], effective masses of electron and hole [33], the dielectric permittivity value [33], the refractive index [33], and the conduction and valence band offsets of 70% and 30%, respectively [4], were considered. A representative electric field value of 2.5 × 10^6^ V/m was assumed for the calculations, which can be obtained for typical CZTSSe solar cells by the Poisson equation [23]. An input power of 100 mW/cm^2^ from the AM1.5G is also considered. Wolfram Mathematica 6.0 software was used as a tool for calculations. Particularly, a package containing all equations, constants, and data was previously developed for the numerical evaluation of quantum well solar cells. This package has allowed the simulation of different types of quantum well solar cells based on AlGaAs, GaInNAs, and SnSSe materials [18,19,20,30].

## 3. Results and Discussion

The potential use of Cu_2_ZnSn(S,Se)_4_ material in solar cells is evaluated in a first step under the radiative limit conditions and without the inclusion of wells. The optoelectronic parameters under ideal conditions are estimated so that they can be compared to the ones obtained when quantum wells are incorporated into the kesterite material. Under ideal conditions, material composition and thickness are two crucial parameters to be considered since different compositions result in a variable band-gap while material thickness controls photon absorption. In this sense, results on the potential application of Cu_2_ZnSn(S,Se)_4_ material into an ideal solar cell are presented in Figure 2. Efficiency values ranging from 13.15 to 29.7% with Jsc and Voc values in the ranges of 19.3–48.4 mA/cm^2^ and 0.628–1.16 V, respectively, are observed in Figure 2a–c. In particular, the lowest Jsc values are obtained for an S/(S + Se) composition near 1 (CZTS material) and very thin layers as illustrated in Figure 2b, due to lower absorption of photons. Therefore, the highest Jsc values are illustrated for an S/(S + Se) composition of 0 (CZTSe material) and the biggest thicknesses as lower band-gap and higher thicknesses promote device absorption. The opposite trend is found for Voc (Figure 2c), for which the greatest values are achieved at the lowest thicknesses and an S/(S + Se) composition near 1 (CZTS material) since higher band-gap values and lower thicknesses result in reduced carrier losses. Efficiency values are optimized for an S/(S + Se) compositional ratio around 0.7 for thicknesses ranging from 1 to 2.5 µm as observed in Figure 2a. In particular, the maximum efficiency expected for CZTSSe without the addition of quantum wells is 29.7%, with Voc and Jsc values of 990 mV and 33.85 mA/cm^2^, respectively, for a CZTSSe thickness of 2 µm and an S/(S + Se) compositional ratio of 0.74. Therefore, a CZTSSe thickness of 2 µm can be considered for further calculations since it allows for almost complete photon absorption. On the other hand, for a CZTSSe thickness of 2 µm, Jsc values of 48.2 mA/cm^2^ and 29.0 mA/cm^2^, Voc values of 658 mV and 1.11 V, and efficiencies of 26.4% and 28.5% are expected for CZTSe and CZTS solar cells, respectively. The maximum values obtained for solar cells without nanostructures are similar to the ones calculated under the Shockley–Queisser approach [34]. Since the mentioned values of Jsc and Voc are the maximum ones expected for CZTSe and CZTS solar cells, they will be further considered by comparing them with the ones obtained when quantum wells are added into kesterite solar cells.

The success of introducing CZTSSe quantum wells into CZTS material to achieve a promoted solar cell efficiency, depends on the management of the number and energy of quantum states formed within the quantum well. The most important advantage of quantum wells is that quantum well energy states can be tailored depending on nanostructure size. The eigenstate values for electrons and light holes were calculated considering a CZTS barrier material and CZTSe wells with different thicknesses, the results are presented in Figure 3a,b. A well depth of 350 meV is characterized by 14 energy levels for electrons and 10 energy levels for light holes when thickness is increased to 50 nm as shown in Figure 3a,b. Therefore, the presence of discretized levels is still observed for kesterite quantum wells with a thickness of 50 nm due to quantum confinement. The number and position of energy levels depends on the quantum well thickness. A general trend is observed in Figure 3 where the number of energy levels increases while level position shifts to lower energies with quantum well thickness increasing to 50 nm, which is expected based on quantum mechanics theory. Consequently, for the lowest thicknesses only a few energy levels will be playing a fundamental role in carrier generation and recombination processes.

The addition of CZTSe quantum wells into CZTS material of 2 µm width is evaluated in the next step, where the number of quantum wells is changed from 20 to 100 wells, keeping a well thickness of 15 nm constant. The well number was chosen considering that a minimum barrier of 5 nm width is required to neglect the effect of the wave functions overlapping between wells and, therefore, there is no interaction between wells. The J-V characteristics depending on the quantum well number are shown in Figure 4. Efficiency increases as well number increases, which can be explained by the increase in Jsc with higher well numbers due to quantum wells allowing for extra photon absorption. That is, CZTS/CZTSe quantum wells result in the formation of discrete levels with energies between the CZTSe and CZTS band-gaps as illustrated in Figure 3, playing a fundamental role in increasing photon absorption in comparison to traditional CZTS bulk material. On the other hand, a slight drop in Voc is illustrated in Figure 4 with the increase in well number—which does not affect the final trend of efficiency—as a result of the increase in carrier recombination due to quantum confinement in wells. Consequently, results presented in Figure 4 are quite promising since the main goal of having nanostructures that favor higher generation rates than recombination rates is being fulfilled.

The trends of efficiency, Jsc, Voc, and J_0_ with quantum well number are shown in Figure 5. The well number varied from 1 to 100, while well thickness was set to 15 nm. Figure 5a illustrates that solar cell efficiency increases from 22% to about 29%, which is mainly a result of the Jsc promotion from 30 to 46 mA/cm^2^ as shown in Figure 5b. The increase in solar cell efficiency with well number increasing have been experimentally reported for other types of solar cells [35,36]. A slight reduction in Voc of about 0.1 V is found in Figure 5c, which is associated with the J_0_ increase with well number enlarging as illustrated in Figure 5d. The increase in quantum well number results in higher confinement of carriers, favoring carrier recombination and thereby increasing J_0_ and reducing Voc. It is important to point out that the optoelectronic parameters of bulk solar cells of 2 µm width with compositions 0 and 1 were added to Figure 5 for comparison. Efficiencies higher than that of the CZTS bulk solar cell are achieved for a well number greater than 60. However, the maximum efficiency under this condition is not able to overcome the one obtained for the optimized bulk solar cell (composition of 0.74), as previously discussed in Figure 2. On the other hand, Figure 5b shows that when a single well is used, Jsc value is near that of CZTS bulk device with the same thickness, since a single well does not provide significant extra photon absorption to the device. Though the increase in well number up to 100 enhances Jsc, this value is still lower than that of bulk CZTSe of 2 µm, which is due to the maximum absorption of photons occurring at the bulk material in comparison to the nanostructured material, where quantized states are available playing a significant role in the absorption process. In CZTS/CZTSe quantum well solar cells, the consideration of CZTS barrier material yields Voc values higher than those obtained for CZTSe cell, as illustrated in Figure 5c. Nevertheless, the introduction of wells is translated into a Voc drop higher than 0.25 V in comparison to CZTS bulk solar cell since carriers confined at wells are more likely to recombine. In this sense, the results of Figure 5 demonstrate that the inclusion of a higher number of CZTSe wells into CZTS bulk material is not only able to increase Jsc of the device compared to traditional CZTS solar cells but it also causes Voc reduction to values below the ones of traditional bulk CZTS solar cell. Therefore, conditions that optimize the trade-off between these two general trends in quantum well solar cells are further required.

Quantum well thickness is among the most relevant parameters controlling photon absorption and carrier recombination. Its influence on solar cell parameters is now estimated considering well thicknesses of 3, 6, 9, 12, and 15 nm; results are given in Figure 6. For calculations, 100 wells were considered. An increase in Jsc is found for thickness enlarging to 15 nm. The number of quantum states within wells increases with well thickness enlarging, as illustrated in Figure 3, which would increase photon absorption and thereby photocurrent density of the device. On the other hand, Voc is reduced due to the carrier recombination increase. However, the increase in Jsc has a predominant role, causing an increase in device performance as illustrated in the inset of Figure 6. It is interesting to note that similar efficiencies of 28.5% and 28.9% are obtained for the optimal CZTS solar cell without nanostructures (Figure 2) and CZTS/CZTSe quantum well solar cells (inset of Figure 6), respectively. However, J-V characteristics of both devices are different since values of 1.11 V and 29.0 mA/cm^2^ for Voc and Jsc, respectively, were found for the optimal CZTS solar cell without quantum wells, as discussed in Figure 2, while for CZTS/CZTSe quantum well solar cells values of 740 mV and 45.9 mA/cm^2^ are achieved as illustrated in the inset of Figure 6. In other words, the introduction of quantum wells allowed an increased Jsc compared to the bulk device without nanostructures due to higher photon absorption; at the same time, Voc is reduced compared to the bulk device without quantum wells due to higher carrier recombination at wells. As a result of the trade-off between Jsc and Voc, similar efficiencies are obtained. 

A more detailed analysis on the influence of well thickness on solar cell parameters is shown in Figure 7, where values corresponding to devices without nanostructures are added for comparison. The variation of well thickness considering a step of 1 nm results in the finding that efficiency is reduced when well thickness is increased up to 4 nm as shown in Figure 7a. Despite Jsc being enlarged, a more drastic drop in Voc is found for this range of well thickness (1–4 nm) due to J_0_ exponentially increasing, as presented in Figure 7c,d. Consequently, solar cell efficiency is degraded. In other words, gain due to photon absorption is lower than carrier recombination losses when introducing CZTS/CZTSe quantum wells with thicknesses lower than 4 nm. For a well thickness higher than 4 nm, the increase in Jsc is still present in the device (Figure 7b) due to the increment of quantum states withing the wells (Figure 3), enlarging photon absorption. However, it is interesting to note that since J_0_ tends to saturate for higher well thicknesses (Figure 7d), there is a small variation in Voc and subsequently efficiency is increased. Higher well thicknesses favor higher photon absorption at the same time that carriers are assisted by the electric field escaping from wells, hence reducing the localization of carriers and their recombination. The comparison of solar cells with and without quantum wells shows that for narrow wells of few nanometers, Jsc and Voc are quite near the ones obtained for bulk solar cells without wells. Carrier absorption in solar cells with narrow wells is slightly higher than that of the bulk solar cells since there is only a single quantum level as observed in Figure 3. Due to the fact that carrier confinement in narrow wells increases carrier recombination, Voc for quantum well solar cells is lower than the one obtained for the bulk device with the barrier composition but without nanostructures. The increase in well thickness up to 15 nm produces higher values of Jsc and shorter values of Voc. However, a Jsc value lower than that of CZTSe solar cell is found, while greater values of Voc are achieved. The best solar cell efficiency is achieved for well thicknesses of 15 nm. Despite this, efficiency is higher than that of CZTS and CZTSe bulk solar cells, and it does not surpass that of the optimized CZTSSe device without quantum wells.

So far, CZTSe quantum well number and thickness parameters have shown dominant roles in solar cell characteristics. The simultaneous influence of both parameters on solar cell characteristics is analyzed in Figure 8. Efficiency is almost independent of well number for a well thickness lower than 4 nm, as observed in Figure 8a, since at particular thicknesses lower than 4 nm, Voc and Jsc are nearly the same with well number variation. For narrow wells with thicknesses lower than 4 nm, there is only a single energy level (Figure 3) limiting carrier absorption/recombination, and consequently, the variation of well number from 1 to 100 does not provide significant change to the results. In fact, Figure 8b shows that the lowest Jsc is obtained for narrow quantum wells regardless of well number. A similar result is also observed when few wells are added, for which Jsc is almost independent of well width, emphasizing that bulk solar cell characteristics are the dominant ones. On the contrary, Jsc is increased for thicknesses higher than 4 nm, implying that wells start playing a relevant role in photon absorption. Despite a drastic Voc drop being found for well thickness enlarging up to 4 nm, for thicknesses higher than 4 nm, Voc is almost constant since carriers are assisted by the electric field escaping from wells. Figure 8 shows that the best Jsc is obtained for the highest well number and thickness, while the greatest Voc is observed for narrow wells due to the best Voc values being obtained for the CZTS solar cell without nanostructures. As a result of these trends, the best efficiency values are not only seen for quantum wells with the highest number and thicknesses but also for very narrow wells, this value being about 29%, which is lower than that of the optimized CZTSSe device without wells. In other words, it is important to explore the use of higher well numbers and thicknesses to break up the efficiency barrier of 30%. Another important point to be highlighted from Figure 8 is that well thickness changes produce higher Jsc and Voc variation in comparison to well number changes. Therefore, photon absorption and carrier recombination are more influenced by the variation of well thickness rather than well number.

Results of Figure 8 demonstrated that higher well thickness and number are required for the best efficiencies. However, it was shown that well thickness plays the most relevant role due to the incorporation of new energy levels at wells. In this sense, for the next step, well thickness was increased from 15 nm up to 50 nm as previously discussed in Figure 3, as it is necessary to reduce the well number to keep barrier thicknesses greater than or equal to 5 nm. Results on the impact of different well thickness and number on the optoelectronic parameters are presented in Figure 9. The increase in well thickness results in higher Jsc and Voc values. Higher thicknesses allow more photon absorption while carrier confinement at wells is reduced, facilitating their escape to the bulk material. It is important to point out that efficiencies of solar cells with nanostructures are greater than that of devices without nanostructures for well width values greater than 20 nm. The highest performance of 36.2% is obtained when 36 wells of 50 nm are embedded in the CZTS material of 2 µm. In particular, for this maximum efficiency point, values of 48 mA/cm^2^ and 867 mV are illustrated for Jsc and Voc, respectively. A Jsc value of 48 mA/cm^2^ is slightly shorter than that of the CZTSe bulk device without nanostructures, as previously discussed in Figure 2. Consequently, a CZTSe well with 50 nm width almost guarantees the complete absorption of photons. On the other hand, a Voc value of 867 mV is between that of the optimized CZTSe (658 mV) and CZTS (1.11 V) devices, as presented in Figure 2. Hence, adding 36 wells of 50 nm width allows for the increase in Jsc, this gain being higher than losses due to Voc reduction, thereby resulting in an efficiency promotion to 36.2%. Therefore, for the next calculations 36 wells of 50 nm width will be further considered. In other types of quantum well solar cells, experimental results have shown that the addition of more than 50 wells results in higher interface recombination [22,35]. Consequently, the use of 36 wells could be more appropriate from the experimental viewpoint.

So far in our analyses, the optimization process has only considered wells based on CZTSe material (S/(S + Se) = 0). The partial replacement of Se atoms by S in the CZTSe material increases its band-gap and thereby well depth is reduced. Despite the fact that a reduction in well depth might result in less photon absorption, carrier recombination is decreased thereby increasing Voc. Therefore, S/(S + Se) composition influence on solar cell characteristics should be studied in detail. The influence of different S/(S + Se) compositions on solar cells is presented in Figure 10. Since with the increase in well composition the well depth is reduced, less photons are absorbed and consequently Jsc is shortened. However, the opposite trend is observed with the Voc, where higher S/(S + Se) compositional ratios at wells favor the increase in Voc due to shallow well formation. Figure 10 also illustrates that an efficiency promotion is further possible. Particularly, the inset of Figure 10 shows that an efficiency of 37.2% is expected for an S/(S + Se) composition of 0.2.

A detailed analysis on S/(S + Se) well composition—from 0 to 0.99—influence on solar cells is illustrated in Figure 11a–d. For calculations, 36 wells of 50 nm width embedded in the CZTS material of 2 µm are considered. Furthermore, the optoelectronic parameters corresponding to CZTS (x = 1) and CZTSe (x = 0) devices without nanostructures are added for comparison. A reduction in Jsc is found when S/(S + Se) composition is increased (Figure 11b). When composition at wells is near zero—wells based on CZTSe material—Jsc is very close to that of the optimized CZTSe device without wells since the complete absorption of photons is almost achieved as discussed previously in Figure 9b. However, when composition at wells is near 1—wells are so shallow that the effect of nanostructures can be neglected—Jsc is similar to the one obtained for the optimized CZTS device without nanostructures, which is an expected result. On the other hand, when S/(S + Se) composition is increased up to 0.55, a lineal increase in Voc is found (Figure 11c), which is a result of the lineal J_0_ reduction (Figure 11d). For S/(S + Se) well compositions higher than 0.55, since carriers easily escape with the help of the electric field from wells, Voc is close to that of the bulk CZTS device without the inclusion of nanostructures. These trends in Jsc and Voc result in a particular efficiency behavior. When well composition is enlarged, solar cell efficiency is promoted, achieving a maximum value of 37.5% at a well composition of 0.26 as shown in Figure 11a. For compositions higher than 0.26, efficiency is decreased due to a Voc increase which is not able to overcome Jsc losses. It is interesting to note that for all compositional ratios, solar cell efficiencies are bigger than the ones obtained for the optimized CZTS and CZTSe solar cells without nanostructures.

Finally, the simultaneous effect of well and barrier compositions on efficiency, Jsc, and Voc of quantum well solar cells based on CZTSSe material was studied, and the results are shown in Figure 12a–c. At a particular barrier composition, the same trend found in Figure 11 for Jsc is observed. That is, the increase in well composition decreases Jsc due to the higher well composition resulting in a lower well depth. A maximum Jsc of 48.25 mA/cm^2^ is obtained, for which, when comparing this result to the one presented in Figure 2, it can be concluded that almost the complete absorption of photons took place. When it comes to Voc (Figure 12c), this parameter is increased with higher well compositions for barrier compositions higher than 0.7. For barrier compositions lower than 0.7, Voc is nearly constant, with well composition variation being due to the formation of shallow wells favoring carrier escape. An interesting result concerning the role of barrier compositional ratio on Jsc and Voc is observed in Figure 12. At a particular well composition, Jsc is independent of barrier composition, this result can be explained considering the opposite effect of well depth and barrier heigh decrement on Jsc. That is, at a particular well composition, the reduction in barrier height is translated into shallow wells and consequently photon absorption within wells is reduced. However, at the same time, lower barrier height guarantees higher absorption at bulk material due to the bulk band-gap drop. Consequently, there is a compensation effect, which results in nearly constant Jsc values for barrier composition variation. Since the highest Voc is obtained for well and barrier compositional ratios near to 1—bulk material with a negligible effect of nanostructures—and the biggest Jsc is obtained for well composition near to 0—the deepest wells—CZTSSe quantum well solar cell efficiency is optimized for barrier compositional ratios higher than 0.7 and well compositional ratios between 0.2 and 0.4 as displayed in Figure 12a. Particularly, a 37.5% record efficiency is reached for a barrier compositional ratio of 1 and well compositional ratio of 0.25, with values of 43.4 mA/cm^2^ and 981 mV for Jsc and Voc, respectively. These Jsc and Voc are not the maximum ones expected for this technology as illustrated in Figure 12. Hence, further strategies for improving photon absorption and carrier transport are open research to attain the maximum values, thereby promoting solar cell efficiency to the next level. In short, the incorporation of 36 wells of 50 nm width—based on CZTSSe material with S/(S + Se) compositional ratio of 0.25—within CZTS barrier material of 2 µm can potentially enhance device efficiency to values as high as 37.5%. For the optimized CZTSSe solar cell without nanostructures, values of efficiency, Voc, and Jsc of 29.7%, 990 mV, and 33.85 mA/cm^2^, respectively, are expected for an S/(S + Se) compositional ratio of 0.74 and a CZTSSe thickness of 2 µm. Consequently, the addition of CZTSSe quantum wells into CZTS solar cells can enhance photocurrent density by a value of about 10 mA/cm^2^, with Voc being slightly lower than the value corresponding to the optimized bulk CZTSSe solar cell, and therefore efficiency is enhanced from 29.7% to 37.5%.

It is important to remark that the superiority of CZTS/CZTSSe quantum well solar cells is demonstrated when comparing the results of this work to the ones of other technologies such as AlGaAs/GaAs and SnS/SnSSe quantum well solar cells. In the former case of AlGaAs/GaAs quantum well solar cells, efficiencies lower than 30% are expected [37], since photon absorption is limited to energies higher than GaAs bulk band-gap of 1.4 eV, unlike CZTS/CZTSSe quantum well solar cells, which allow the absorption of photons with energy higher than CZTSe bulk band-gap of 1.0 eV. Consequently, higher Jsc values are expected in CZTS/CZTSSe quantum well solar cells, which can result in an efficiency value of 37.5%. The advantage of CZTSSe quantum wells is also demonstrated when comparing the results of this work to the ones obtained for SnS solar cells with an SnSSe quantum well, for which, well thicknesses of 54 nm can result in a maximum efficiency of 32.1%, with values of 40.6 mA/cm^2^ and 906 mV for Jsc and Voc, respectively [30]. When it comes to SnS/SnSSe quantum well solar cells, the addition of SnSe quantum wells would only guarantee the extra photon absorption in the range of 1.0 to 1.3 eV, compared to the range of 1.0 to 1.5 eV when using CZTSSe quantum wells. 

The radiative limit provides not only information on the maximum values expected for the optoelectronic parameters of the analyzed device, but also shows the fundamental requirements to be attained for achieving the maximum values. To achieve the best performances in CZTS/CZTSSe quantum well solar cells at the lab level, a crystalline quality of materials and their coupling should guarantee lesser carrier transport losses. The implementation of quantum wells into kesterite solar cells should be developed considering a p-i-n structure. That is, traditional substrates of Mo can be used to deposit in a first step a typical CZTS material with p-type conductivity, followed by an intrinsic region deposition consisting of sandwiched layers of CZTS and CZTSSe, to finally complete the device fabrication with the deposition of traditional thin films such as CdS buffer layer (n-type semiconductor) as well as ZnO and ZnO:Al window materials. Traditional techniques for thin film deposition can be used to obtain Mo, CZTS with p-type conductivity, CdS, ZnO, and ZnO:Al. However, since it is necessary to obtain a precise well and barrier thickness and composition control during film growth of the intrinsic region, traditional deposition techniques cannot be used for the intrinsic region deposition. The Molecular Beam Epitaxy (MBE) is an adequate technique for the fine processing of these structures. In fact, MBE is a potential technique to obtain a single-phase with high crystalline quality in layers as is required for high-efficiency solar cells. Despite there only being a few works exploring kesterite thin film deposition by MBE in the literature, deposition rates of about 1 Å/s have been informed during kesterite layer deposition [38,39], which make the fabrication of devices with the required composition and with very thin wells and barriers possible. From the technological point of view, this is an important result since very thin barriers of 5 nm could be processed within moderate times of 50 s. Nevertheless, the processing of CZTSSe by MBE is just in an early stage, with further studies being necessary to obtain pure phase kesterite material with the required crystalline quality for device application. The attenuation of current issues concerning kesterite solar cells such as bulk defects, interface defects, grain boundaries, and secondary phases is a prior mandatory step for the success of kesterite quantum well solar cells. While MBE is quite an attractive technique to obtain the required crystalline quality in well and barrier materials, the replacement of CdS is highly recommended to reduce kesterite/buffer interface defects. In this sense, the use of CdZnS could be a more appropriate proposal to not only reduce the interface defects but also to reduce the Cd concentration while allowing long wavelength photon absorption due to the increase in buffer layer band-gap in comparison to the traditional CdS layer [40]. These points are open research. 

## 4. Summary

In summary, CZTS/CZTSSe quantum well solar cells are introduced as a potential route for efficiency promotion. In particular, an analytical study under the radiative limit was presented. The physics governing absorption and recombination processes was analyzed for solar cells with and without quantum wells. The results showed that our proposal allows for broadening quantum efficiency due to discrete eigenstates formed at wells, thereby contributing to the Jsc increase in solar cells. As a general tendency, the incorporation of quantum wells results in Voc reduction. Therefore, it was important to find conditions which result in an efficiency greater than that of a device without nanostructures. We found that thin wells are not able to obtain efficiencies higher than that of the bulk device due to higher carrier confinement at wells. Only for well thicknesses higher than 20 nm, are efficiency values higher than 29% expected. An important result found is that well thickness plays a more fundamental role in the optoelectronic parameters rather than well number. After the optimization of CZTS/CZTSSe quantum well solar cells, we found that 36 wells with a thickness of 50 nm, characterized by an S/(S + Se) well compositional ratio of 0.25 can result in an efficiency value of 37.5%. However, in order to achieve this goal, further studies are needed to refine the kesterite deposition process through the MBE technique so that the required material crystalline quality, thickness, and composition can be attained.

## Figures and Tables

**Figure 1 nanomaterials-13-02058-f001:**
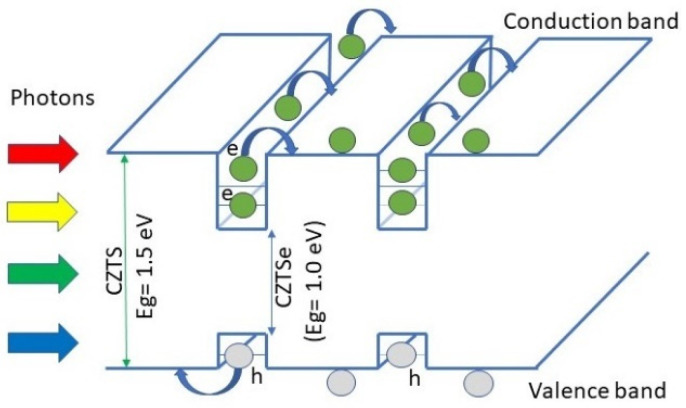
Quantum well solar cell based on CZTSSe material.

**Figure 2 nanomaterials-13-02058-f002:**
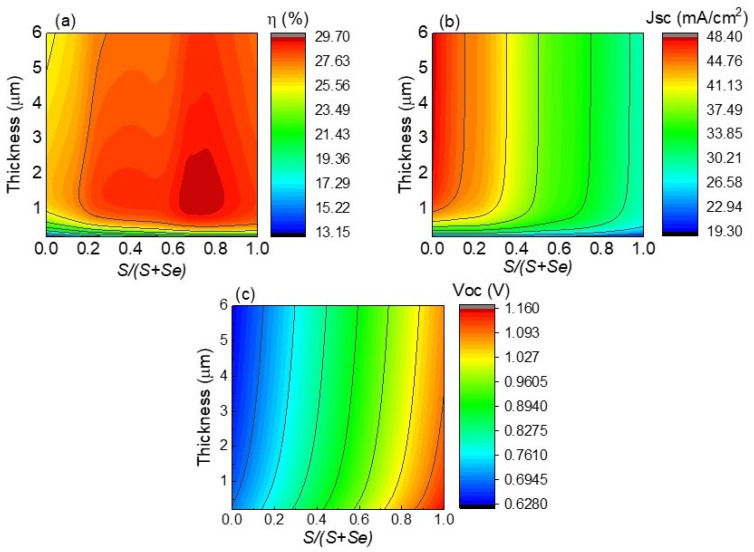
Efficiency (**a**), Jsc (**b**), and Voc (**c**) of CZTSSe solar cells as functions of thickness and composition.

**Figure 3 nanomaterials-13-02058-f003:**
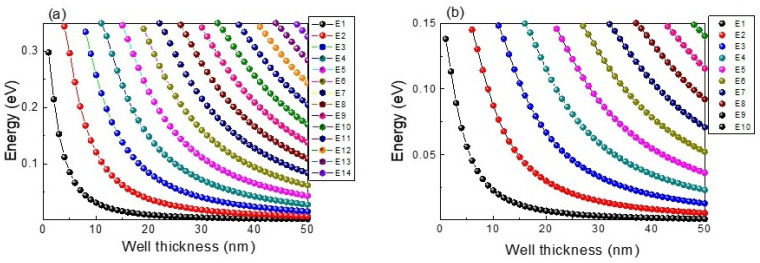
Eigenstate values in CZTS/CZTSe quantum wells for electrons (**a**) and light holes (**b**) as functions of well thickness.

**Figure 4 nanomaterials-13-02058-f004:**
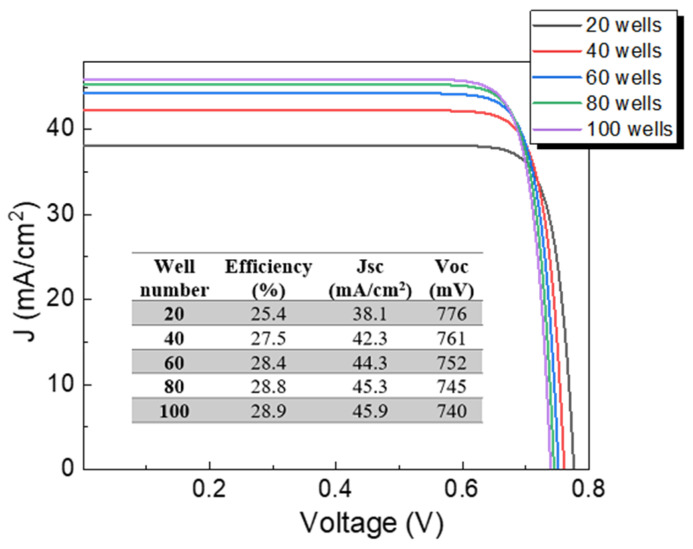
J-V characteristics of CZTS solar cells with CZTSe quantum wells varying well number, considering a fixed well thickness of 15 nm. The inset shows the optoelectronic parameters of devices.

**Figure 5 nanomaterials-13-02058-f005:**
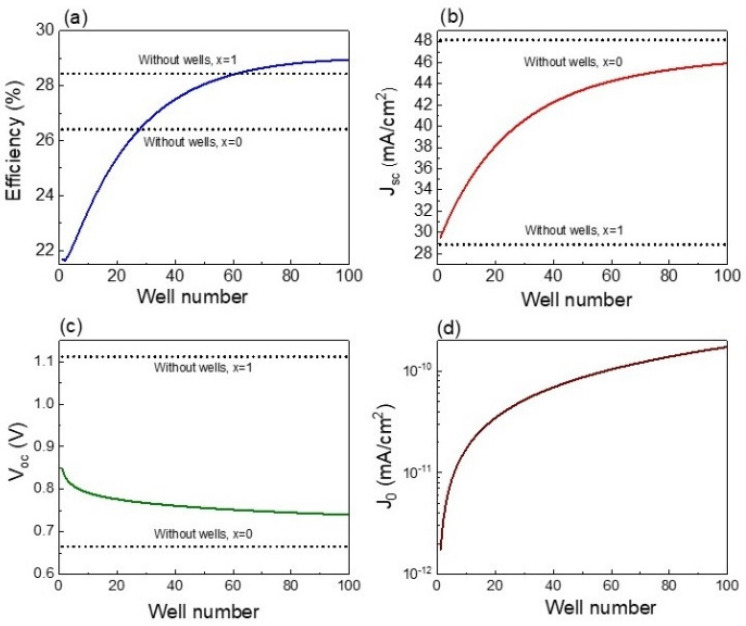
Efficiency (**a**), Jsc (**b**), Voc (**c**), and J_0_ (**d**) as functions of well number. The optoelectronic parameters corresponding to CZTS (x = 1) and CZTSe (x = 0) devices without nanostructures are added for comparison.

**Figure 6 nanomaterials-13-02058-f006:**
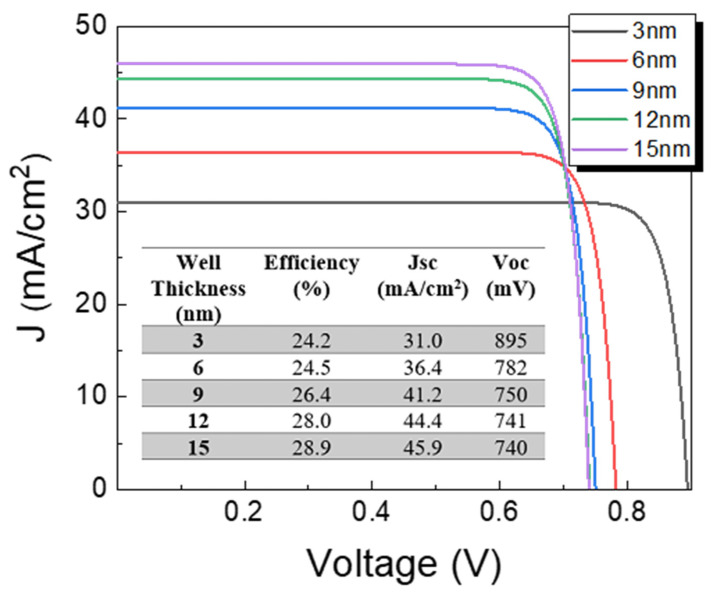
J-V characteristics of solar cells with CZTS/CZTSe quantum wells, varying well thickness for a fixed well number of 100. The inset shows the optoelectronic parameters of devices.

**Figure 7 nanomaterials-13-02058-f007:**
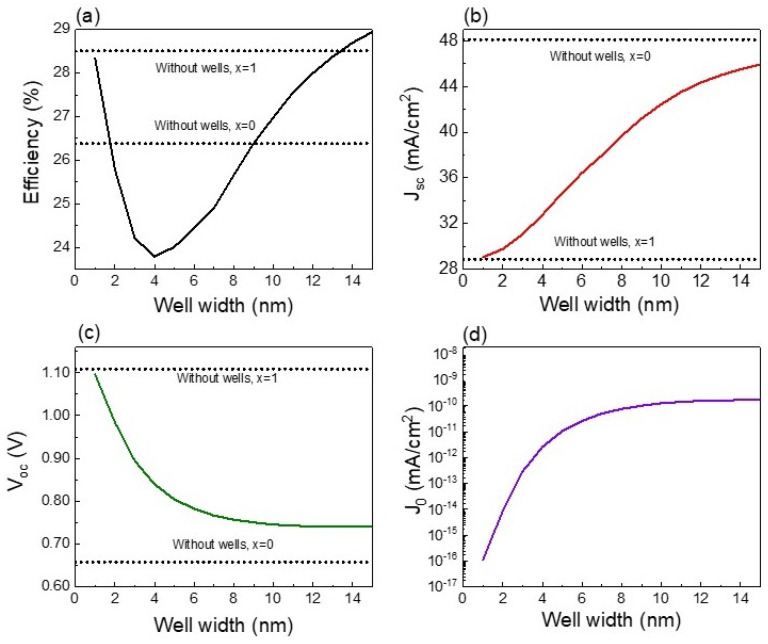
Efficiency (**a**), Jsc (**b**), Voc (**c**), and J_0_ (**d**) as functions of well thickness. The optoelectronic parameters corresponding to CZTS (x = 1) and CZTSe (x = 0) devices without nanostructures are added for comparison.

**Figure 8 nanomaterials-13-02058-f008:**
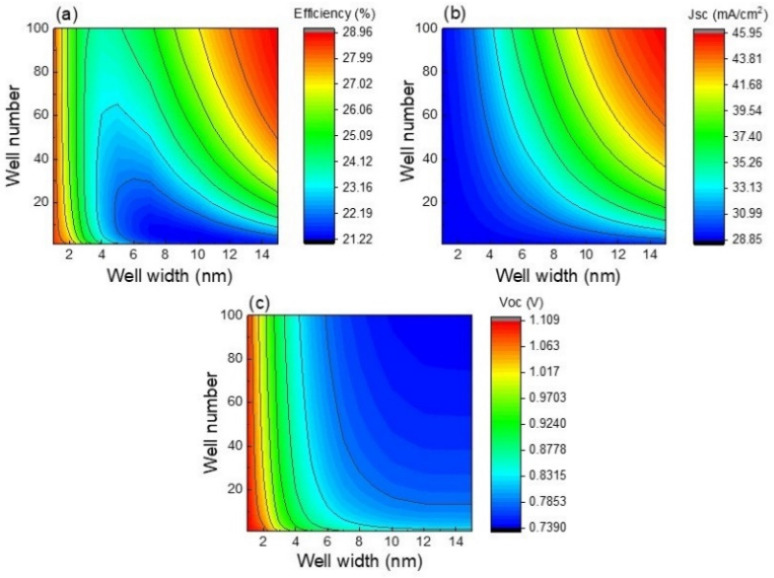
Efficiency (**a**), Jsc (**b**), and Voc (**c**) as functions of well thickness and number.

**Figure 9 nanomaterials-13-02058-f009:**
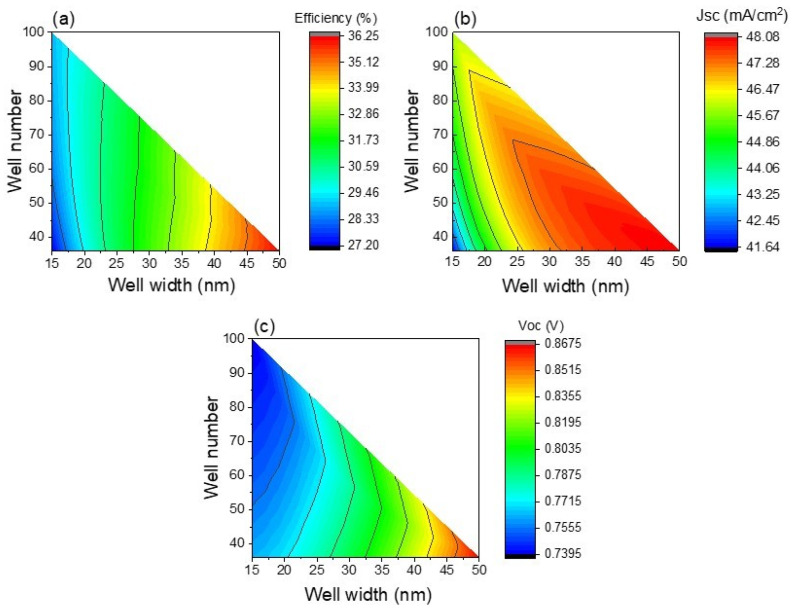
Efficiency (**a**), Jsc (**b**), and Voc (**c**) as functions of well thickness and number, where well thicknesses higher than 15 nm are evaluated.

**Figure 10 nanomaterials-13-02058-f010:**
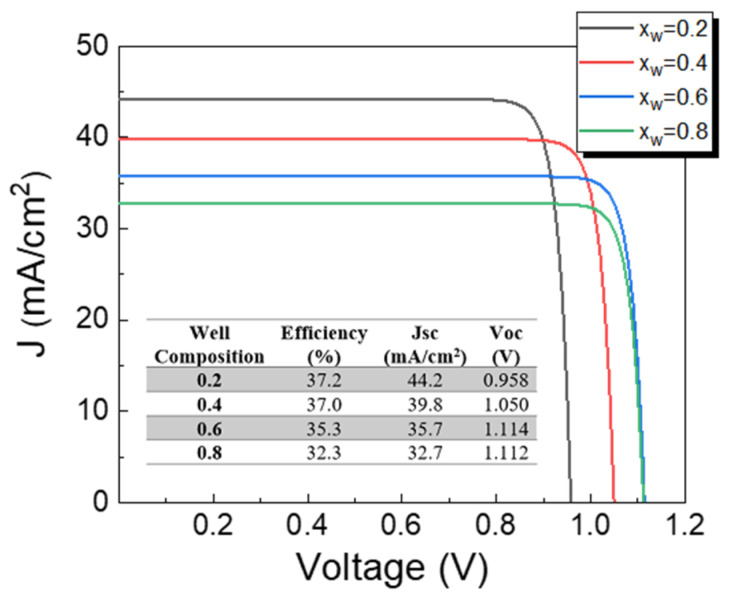
Influence of different S/(S + Se) compositions at wells (0.2, 0.4, 0.6, and 0.8) on a CZTS solar cell with CZTSSe quantum wells. The inset shows the optoelectronic parameters of devices.

**Figure 11 nanomaterials-13-02058-f011:**
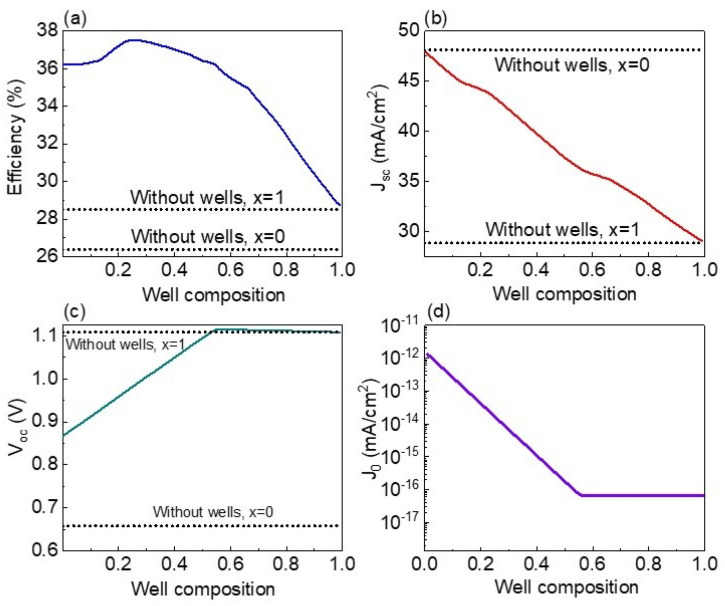
Efficiency (**a**), Jsc (**b**), Voc (**c**), and J_0_ (**d**) as functions of well composition (S/(S + Se) compositional ratio). The optoelectronic parameters corresponding to CZTS (x = 1) and CZTSe (x = 0) devices without nanostructures are added for comparison.

**Figure 12 nanomaterials-13-02058-f012:**
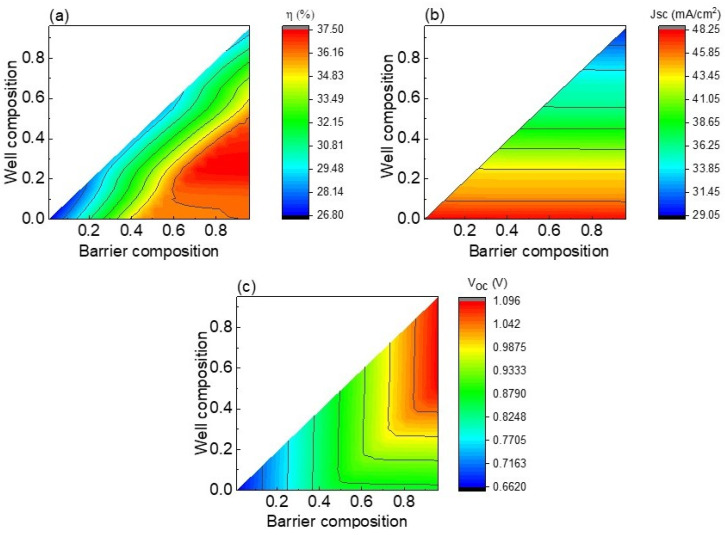
Efficiency (**a**), Jsc (**b**), and Voc (**c**) as functions of well and barrier compositions.

## Data Availability

Not applicable.

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
