# Peer review of "Analytical Modeling and Optimization of Cu2ZnSn(S,Se)4 Solar Cells with the Use of Quantum Wells under the Radiative Limit"

_nanomaterials, 2023, doi:10.3390/nano13142058_

Round 1

Reviewer 1 Report

In this work, the authors present a theoretical study on the use of Cu2ZnSn(S,Se)4 quantum wells 24 into Cu2ZnSnS4 solar cells to enhance device efficiency. The content is significant and presented well. It can be accepted after a minor revision.

1. For Figure 6, J-V characteristics of solar cells with CZTS/CZTSe quantum well. Dose it means J-V characteristics of solar cells with CZTS quantum well is same with J-V characteristics of solar cells with CZTSe quantum well?

2. The inset of Figure 4, Figure 6 and Figure 10 are very fuzzy, it should be presented larger and clearer.

Author Response

Reviewer #1

General comments:

In this work, the authors present a theoretical study on the use of Cu2ZnSn(S,Se)4 quantum wells into Cu2ZnSnS4 solar cells to enhance device efficiency. The content is significant and presented well. It can be accepted after a minor revision.

Dear referee: 

We want to thank you for your fruitful comments and your positive feedback. All of your suggestions were carefully analyzed and we found them quite interesting for manuscript improvement. Below it is added a response letter for each outlined point in which your commentaries are in bold. All corrections added to the manuscript were highlighted in red color. We hope to submit a superior paper with your indications which will be published on Nanomaterials.

Suggestions and responses: 

  1. Figure 6, J-V characteristics of solar cells with CZTS/CZTSe quantum well. Does it mean J-V characteristics of solar cells with CZTS quantum well is same with J-V characteristics of solar cells with CZTSe quantum well?

Answer:

Thank you very much for your comment. Please note that in the example you kindly mentioned of CZTS quantum wells addition into CZTS bulk material it means that we are dealing with bulk CZTS solar cell – CZTS without nanostructures.  Therefore, we think the reviewer #1 is actually concerned about the similarities between J-V characteristics of CZTS solar cells with CZTSe quantum wells and the J-V characteristics of CZTS bulk material. As a matter of fact, similar efficiencies of 28.5% and 28.9% are obtained for CZTS solar cell without nanostructures and CZTS/CZTSe quantum well solar cells, respectively. However, please note that J-V characteristics of both devices are different, which can be observed from the values of Jsc and Voc of both devices. In particular, for CZTS bulk solar cell – CZTS solar cell with CZTS quantum wells – values of 1.11 V and 29.0 mA/cm2 for Voc and Jsc, respectively, are reported in the manuscript as shown in Figure 2, while for CZTS solar cell with CZTSe quantum wells, Voc and Jsc values of 740 mV and 45.9 mA/cm2 are obtained as illustrated in the inset of Figure 6. That is, the introduction of quantum wells it results in an increased Jsc compared to the bulk device without nanostructures due to higher photon absorption, at the same time Voc is reduced compared to the bulk device without nanostructures due to higher carrier recombination at wells. Consequently, similar efficiencies are obtained as claimed by the reviewer #1. In order to clarify this point, the following sentences were added to this new manuscript version. Please find the following new text in the manuscript:

New text at page 9 of the new manuscript version:

However, the increase in Jsc has a predominant role, causing an increase in device performance as illustrated in the inset of Figure 6. It is interesting to note that similar efficiencies of 28.5% and 28.9% are obtained for optimal CZTS solar cell without nanostructures (Figure 2) and CZTS/CZTSe quantum well solar cells (inset of Figure 6), respectively. However, J-V characteristics of both devices are different since values of 1.11 V and 29.0 mA/cm2 for Voc and Jsc, respectively, were found for optimal CZTS solar cell without quantum wells, as discussed in Figure 2, while for CZTS/CZTSe quantum well solar cells values of 740 mV and 45.9 mA/cm2 are achieved as illustrated in the inset of Figure 6. In other words, the introduction of quantum wells it allowed an increased Jsc compared to the bulk device without nanostructures due to higher photon absorption; at the same time, Voc is reduced compared to the bulk device without quantum wells due to higher carrier recombination at wells. As a result of the trade-off between Jsc and Voc, similar efficiencies are obtained.”

  1. The inset of Figure 4, Figure 6 and Figure 10 are very fuzzy, it should be presented larger and clearer.

Answer:

Thank you very much for pointing this out. We have just realized that there was an error while converting the figures from the word document to pdf format, where the tables of inset turned to be fuzzy. We apology for this mistake and the inset of figures are now with the adequate quality.   

Once again, we thank you for your kind recommendations and we hope to submit a superior paper with all referees’ recommendations which will be able to be published on Nanomaterials.

Reviewer 2 Report

The author proposed a very good topic and seems interesting, though the applicability of such quantum wells looks quite difficult.

1. Can the author highlight more about the possible methodology and practical implementation way of such quantum wells with kesterite?

2.  There are several reasons for the low Voc deficit and possible strategies to reduce them pls mention them, an author can cite and get additional information from the below literature and highlight them in the introduction part. ,  J. Mater. Chem. A, 2022,10, 8466-8478,  ACS Appl. Mater. Interfaces 2021, 13, 1, 429–437

3. It would be great if the author compared the kesterite quantum well system with other materials and efficiency to prove their superiority 

Author Response

Reviewer #2

Dear referee: 

We want to thank you for your fruitful comments and your positive feedback. All of your suggestions were carefully analyzed and we found them quite interesting for manuscript improvement. Below is added a response letter for each outlined point in which your commentaries are in bold. All corrections added to the manuscript were highlighted in red color. We hope to submit a superior paper with your indications which will be published on Nanomaterials. 

General comment

The author proposed a very good topic and seems interesting, though the applicability of such quantum wells looks quite difficult.

Answer: 
Thank you very much for your positive feedback. We have improved the manuscript quality following your kind suggestions. In this manuscript version we have discussed on main challenges concerning this technology as presented below.

  1. Can the author highlight more about the possible methodology and practical implementation way of such quantum wells with kesterite?

Answer: 
Thank you very much for your interesting suggestion. In this new manuscript version, we have added more information concerning the possible methodology and practical implementation of kesterite quantum well solar cells. In particular, we recommend the use of MBE deposition technique for the precise control of well and barrier thicknesses and compositions, while guaranteeing the formation of single-phase and good crystalline quality of layers. We presented the need of a p-i-n structure based on traditional materials such as Mo, p-type CZTS material, CdS, ZnO and ZnO:Al for the fabrication of kesterite quantum well solar cells, highlighting the need of replacing the traditional CdS with CdZnS to reduce interface defects. Please find the following new text in the manuscript:

New text at page 16-17 of the new manuscript version:

The radiative limit provides not only information on the maximum values expected for the optoelectronic parameters of the analyzed device, but also shows the fundamental requirements to be attained for achieving the maximum values. To achieve the best performances in CZTS/CZTSSe quantum well solar cells at lab level, crystalline quality of materials and their coupling should guarantee the lesser carrier transport losses. The implementation of quantum wells into kesterite solar cells should be developed considering a p-i-n structure. That is, traditional substrates of Mo can be used to deposit in a first step a typical CZTS material with p-type conductivity, followed by an intrinsic region deposition consisting of sandwiched layers of CZTS and CZTSSe, to finally complete the device fabrication with the deposition of traditional thin films such as CdS buffer layer (n-type semiconductor) as well as ZnO and ZnO:Al window materials. Traditional techniques for thin film deposition can be used to obtain Mo, CZTS with p-type conductivity, CdS, ZnO and ZnO:Al. However, since it is necessary the precise well and barrier thickness and composition control during films growth of the intrinsic region, traditional deposition techniques cannot be used for the intrinsic region deposition. The Molecular Beam Epitaxy (MBE) is an adequate technique for the fine processing of these structures. In fact, MBE is a potential technique to obtain single-phase with high crystalline quality in layers as required for high efficiency solar cells. Despite there are only few works exploring kesterite thin film deposition by MBE in the literature, deposition rates of about 1 Å/s have been informed during kesterite layer deposition [38, 39], which make possible the fabrication of devices with the required composition and with very thin wells and barriers. From the technological point of view, this is an important result since very thin barriers of 5 nm could be processed within moderate times of 50 s. Nevertheless, the processing of CZTSSe by MBE is just in an early stage, being necessary further studies to obtain pure-phase kesterite material with the required crystalline quality for device application. The attenuation of current issues concerning kesterite solar cells such bulk defects, interface defects, grain boundaries and secondary phases is a prior mandatory step for the succeed of kesterite quantum well solar cells. While MBE is a quite attractive technique to obtain the required crystalline quality in well and barrier materials, the replacement of CdS is highly recommended to reduce kesterite/buffer interface defects. In this sense, the use of CdZnS could be a more appropriate proposal to not only reduce the interface defects but also to reduce the Cd concentration while allowing long wavelength photon absorption due to the increase of buffer layer band-gap in comparison to the traditional CdS layer [40]. These points are open research.

New reference:

Nicolás-Marín, M.M.; Ayala-Mato, F.; Vigil-Galán, O.; Courel, M. Simulation analysis of Cd1-xZnxS/Sb2 (Se1-xSx)3 solar cells with nip structure. Solar Energy 2021, 224, 245-252

  1. There are several reasons for the low Voc deficit and possible strategies to reduce them pls mention them, an author can cite and get additional information from the below literature and highlight them in the introduction part. ,   Mater. Chem. A, 2022,10, 8466-8478,  ACS Appl. Mater. Interfaces2021, 13, 1, 429–437

Answer: 
Thank you very much for your kind recommendation. Following your suggestion, in this manuscript version we have added more information on the reasons of low Voc as well the importance of paying attention to the relationships between structure, composition and electronic properties to enhance Voc. In order to clarify this important point to the reader, the following new text was added to the manuscript:

New text added to the introduction section:

An important problem of CZTSSe solar cell in comparison to technologies such as CIGS is the deficit of open-circuit voltage (Voc) [3,6,7], while short-circuit current density (Jsc) values are close for both devices [8]. A high concentration of defects along with grain boundaries, defect clusters, and poor band-tailing characteristics have been identified as the main causes of Voc-deficit [9,10]. In particular, it was found that under Cu-poor and Zn-rich conditions, the relative concentration of the ZnCu and SnCu defects and cluster defects such as [2ZnCu + ZnSn] and [2CuZn + SnZn] play a fundamental role in the Voc and thereby on the power conversion efficiency [9]. The relationships between structure, composition and electronic properties have been presented as a key factor for reducing the Voc-deficit in this technology [10]. However, despite effort performed to improve Voc parameter, so far, results on this research line have been discouraging. Consequently, new alternative approaches to improve CZTSSe solar cell efficiency are required.”

New references:

Karade, V. C.; Suryawanshi, M. P.; Jang, J. S.; Gour, K. S.; Jang, S.; Park, J.; Kim, J. H.; Shin, S. W. Understanding defects and band tailing characteristics and their impact on the device performance of Cu2ZnSn(S,Se)4 solar cells. J. Mater. Chem. A 2022, 10, 8466.

Karade, V.; Choi, E.; Gang, M. G.; Hyesun, Y.; Lokhande, A.; Babar, P.; Jang, J. S.; Seidel, J.; Yun, J. S.; Park, J.; Kim, J. H. Achieving Low VOC-deficit Characteristics in Cu2ZnSn(S,Se)4 Solar Cells through Improved Carrier Separation. ACS Appl. Mater. Interfaces 2021, 13, 429−437

  1. It would be great if the author compared the kesterite quantum well system with other materials and efficiency to prove their superiority 

Answer: 
Thank you very much for your nice recommendation. In this manuscript version we have added information on the comparison of our results with the ones obtained in the literature for other technologies such as AlGaAs/GaAs and SnS/SnSSe quantum well solar cells. Based on the photon absorption behavior, we explain why CZTS/CZTSSe quantum well solar cells are better candidates and consequently better results are obtained. In order to highlight the superiority of kesterite quantum well solar cells in comparison to other technologies, the following new sentences can be found in this new manuscript version:

New text at page 16:

It is important to remark that the superiority of CZTS/CZTSSe quantum well solar cells is demonstrated when comparing the results of this work to the ones of other technologies such as AlGaAs/GaAs and SnS/SnSSe quantum well solar cells. In the former case of AlGaAs/GaAs quantum well solar cells, efficiencies lower than 30% are expected [37], since photon absorption is limited to energies higher than GaAs bulk band-gap of 1.4 eV, unlike CZTS/CZTSSe quantum well solar cells, which allow the absorption of photons with energy higher than CZTSe bulk band-gap of 1.0 eV. Consequently, higher Jsc values are expected in CZTS/CZTSSe quantum well solar cells, which can result in an efficiency value of 37.5%. The advantage of CZTSSe quantum wells is also demonstrated when comparing the results of this work to the ones obtained for SnS solar cells with SnSSe quantum well, for which, well thicknesses of 54 nm can result in a maximum efficiency of 32.1%, with values of 40.6 mA/cm2 and 906 mV for Jsc and Voc, respectively [30]. When it comes to SnS/SnSSe quantum well solar cells, the addition of SnSe quantum wells would only guarantee the extra photon absorption in the range of 1.0 to 1.3 eV, compared to the range of 1.0 to 1.5 eV when using CZTSSe quantum wells.

New reference:

Laznek, S.; Meftah, A.; Meftah, A.; Sengouga, N. Semi-Analytical Simulation and Optimization of AlGaAs/GaAs p-i-n Quantum Well Solar Cell. Applied Solar Energy 2018, 54, 261–269.

Once again, we thank you for your kind recommendations and we hope to submit a superior paper with all referees’ recommendations which will be able to be published on Nanomaterials.
